# Machine-Learning Applications in Geosciences: Comparison of Different Algorithms and Vegetation Classes' Importance Ranking in Wildfire Susceptibility

Andrea Trucchia [1,*] , Hamed Izadgoshasb [1] , Sara Isnardi [2] , Paolo Fiorucci [1] and Marj Tonini [3]

1   CIMA Research Foundation, I-17100 Savona, Italy
2   Department of Mathematical Sciences "G.L. Lagrange" Politecnico di Torino, I-10129 Turin, Italy
3   Institute of Earth Surface Dynamics, Faculty of Geosciences and Environment, University of Lausanne, CH-1015 Lausanne, Switzerland
*   Correspondence: andrea.trucchia@cimafoundation.org

**Abstract:** Susceptibility mapping represents a modern tool to support forest protection plans and to address fuel management. With the present work, we continue with a research framework developed in a pioneristic study at the local scale for Liguria (Italy) and recently adapted to the national scale. In these previous works, a random-forest-based modeling workflow was developed to assess susceptibility to wildfires under the influence of a number of environmental predictors. The main novelties and contributions of the present study are: (i) we compared models based on random forest, multi-layer perceptron, and support vector machine, to estimate their prediction capabilities; (ii) we used a more accurate vegetation map as predictor, allowing us to evaluate the impacts of different types of local and neighboring vegetation on wildfires' occurrence; (iii) we improved the selection of the testing dataset, in order to take into account the temporal variability of the burning seasons. Wildfire susceptibility maps were finally created based on the output probabilistic predicted values from the three machine-learning algorithms. As revealed with random forest, vegetation is so far the most important predictor variable; the marginal effect of each type of vegetation was then evaluated and discussed.

**Keywords:** random forest; multi-layer perceptron; support vector machine; vegetation types; partial dependent plot; variable importance ranking; Liguria

## 1. Introduction

Wildfires affect millions of people worldwide and are often characterized by ecosystem and economic impacts at different scales [1]. Forest ecosystems where wildfires are uncommon events, or where non-native vegetation has been introduced, can be negatively affected by wildfires, which can spread to surrounding rural areas, affecting houses and human lives. In such conditions, wildfires constitute a complex environmental disaster triggered by several interacting natural and human factors [2].

Wildfires can severely harm communications, transportation, power and gas services, and water supply. They also have a negative impact on air quality; and lead to loss of property, crops, resources, people, and animals [3]. They also lead in some cases to land degradation: desertification and deforestation constitute examples of the adverse consequences of wildfires [4]. Other impacts of wildfires include damage to wildlife habitats and timber disruption, which can lead to loss of biodiversity and economic damages [5]. In addition to effects on plants and ecosystems, wildfires also affect geological and hydrological processes in the years following the burning. They affect the biosphere, triggering ash deposition, influencing the formation of water-repellent soil, and physically weathering the bedrock. This can lead to erosion through sheetwash, rilling, dry ravel, and increased mass movement, by the means of floods, debris flow, landslides, and rockfall [6].

The Mediterranean basin is particularly prone to wildfires. These events are well documented and cause substantial damages every year [7]. The last annual report of the European Forest Fire Information System [8] records that, in the five large southern European countries (Portugal, Spain, France, Italy, and Greece) between 1980 and 2020, after an increase in the first decade, the number of fires stabilized, and finally, it started decreasing in the last decade. The average fire size followed the same trend, with the exception of the extreme fire season in 2017, which caused a record of 500,000 burned hectares in Portugal [9]. In 2020, the total number of fires was 30,661 and the average fire size was 7 ha, both below the long trend average. This decreasing trend is largely due to the improvement in the fire protection services operating at the local level.

As observed by *Turco and co-authors* [10], fire management is focused mainly on fire suppression. However, climate and land use/land cover changes that affect the ecological and socio-economic vulnerability to wildfires could lead to more complex and unpredictable situations in the future [11,12]. Moreover, fire exclusion and suppression can lead to a higher fuel load and connectivity, allowing a limited long-term positive effect at best [13]. Therefore, fire management strategies should develop and implement prevention and adaptation measures, in addition to the ongoing suppression ones [12,14].

The implementation of accurate wildfire inventories represents a useful tool for tackling prevention and planning programs. These ideally should include the spatial locations of individual events, the size and perimeter of each burned area, the starting and ending dates, and the total durations. Such inventories provide a vital source of information for the development of hazard, risk, and susceptibility maps.

For the sake of clarity, we define "susceptibility of an area" as the potential to experience a particular hazard in the future, based only on the intrinsic local properties of the territory, assessed in terms of relative spatial likelihood [15,16]. "Hazard maps" account also for the potential intensity of the phenomenon, thereby expressing both the likelihood and the likely severity of a wildfire in a given area. Finally, "risk maps" inform us of the damages or losses related to wildfires, by taking into account exposed assets and their vulnerabilities. In the field of wildfire risk management, fire risk is defined as an indicator of the probability that an area will burn in a certain period of time and have destructive impacts on the population and infrastructures [17–20]. The present study focuses on the concept of susceptibility, providing static maps which rely on the assumption that future wildfire events are expected to take place under the same anthropic and geo-environmental conditions as past events. Although these concepts are well consolidated in the risk assessment research area, where landslides constitute a notable example [21], there is a need to expand them to other natural hazards, such as wildfires [15,16,22], and to evaluate different methods for susceptibility assessment and mapping.

In most studies related to wildfire risk mapping, the susceptibility assessment is the first mandatory step to produce further maps [23]. Susceptibility mapping can therefore represent a modern tool to support forest protection plans and to address fuel management strategies in order to reduce fires' consequences [16], and it is a relevant ally for risk reduction programs, land use planning, and civil protection activities.

Several techniques have been tested in the scientific community for risk and susceptibility mapping in relation to natural and anthropogenic hazards. These typically rely on physically based or expert-based models, often integrated into a geographic information system (GIS), or they can include statistical analyses and modeling to assess the relative importance of the predictors [16,24–27]. In particular, purely statistical models have been recently successfully applied in susceptibility mapping for wildfires [28–31]. Multi-criteria decision analysis has been lately applied by Ljubomir et al. [32] to produce a forest fire susceptibility map of an area in the Republic of Serbia. Salavati et al. [33] performed statistical techniques on a set of predisposing factors, such as climatic factors, topographic variables, land use/land cover, and distances from roads and rivers to generate wildfire risk maps. Novo et al. in [34] applied the analytic hierarchy process using several predictors (topographic variables, distances to roads and settlements, vegetation indices and fuel

types, Canadian Fire Weather Index, and historical fire regimes) to produce risk maps at the local level in Galicia (Spain).

The comparison between deterministic methods, both physical and statistical approaches, and methods based on machine learning, highlighted the benefit of using the latter [23,35–37], mainly because of their ability to extract insights from the a huge volume of heterogeneous digital information. In addition, given a set of initial conditions (i.e., wildfire observations and predisposing variables), deterministic methods lead to a unique result, with no way for assessing its uncertainty. On the contrary, in stochastic models (including machine learning), each simulation may give a different result, due to the randomization of the initial conditions, and this way reflects better the reality of complex geo-environmental phenomena. This allows one, in turn, to estimate how sure the model is about its predictions—that is to say, to assess the uncertainty. Recent advances in automated learning and simulation methods, such as machine learning (ML) algorithms, have stimulated great interest in employing intelligent models to create wildfire susceptibility maps in several case studies. Indeed, ML allows one to analyze, model and visualize complex sets of geo-environmental data, and it has performed particularly well in modeling natural hazards, which have intrinsically complex and non-linear behavior [2,15,22,27,38–41].

With the present work, we continue investigating a research framework developed in a pioneering study at the local scale for Liguria region, in Italy [15], and recently adapted to the national scale [16]. In these previous works, a random forest-based ML modeling workflow was developed to generate wildfire susceptibility maps of the study areas, under the influence of a number of environmental predictors (altitude and its derivatives, and vegetation type). At the regional scale, different parameters were tested, and the performances of the related models were evaluated. Including the neighboring vegetation as an additional predictor variable and implementing a 5-fold cross validation procedure allowed us to increase the predictive performance of the model. Due to the heterogeneity of the area at the national scale, climatic variables (namely, the mean temperature and mean precipitation over the last 70 years) were also considered in the second work [16].

The main objective of the present research was to evaluate the importance of different predictors to wildfire susceptibility in a fire-prone area (Liguria), which was assessed by using different ML based approaches. The main novelties and contributions compared to the previous works are the following: (i) we compared three ML algorithms (random forest, multi-layer perceptron, and support vector machine) to assess their prediction capabilities; (ii) we used an accurate and verified vegetation map as the predictor, allowing us to evaluate the impacts of different classes of local and neighboring vegetation on wildfires occurrence; (iii) we improved the selection of the validation dataset in order to take into account the temporal variability of the burning seasons.

This study contributes to filling the gap in the literature concerning: (i) the quantitative assessment of how different classes of vegetation can influence the susceptibility to wildfires, which was achieved directly with an ML model; (ii) the fair and appropriate selection of an independent dataset for model validation, which ensures its ability to perform accurate predictions on new data.

## 2. Material and Methods

### 2.1. Study Area

Liguria is located in the northwest part of Italy. It covers an area of 5400 km$^2$ and is shaped as a narrow strip of land bordered by the Tyrrhenian sea along its entire southern extension (Figure 1). The territory is crossed by the two main Italian mountain ranges: the Alps to the west and the Apennines to the east. Liguria is a mainly mountainous (65%) and hilly (35%) region with no plains and a coast that almost always overlooks the sea. The resulting topography is very complex. The slope is higher than 40% over 50% of the territory, and there is dense and heterogeneous vegetation.

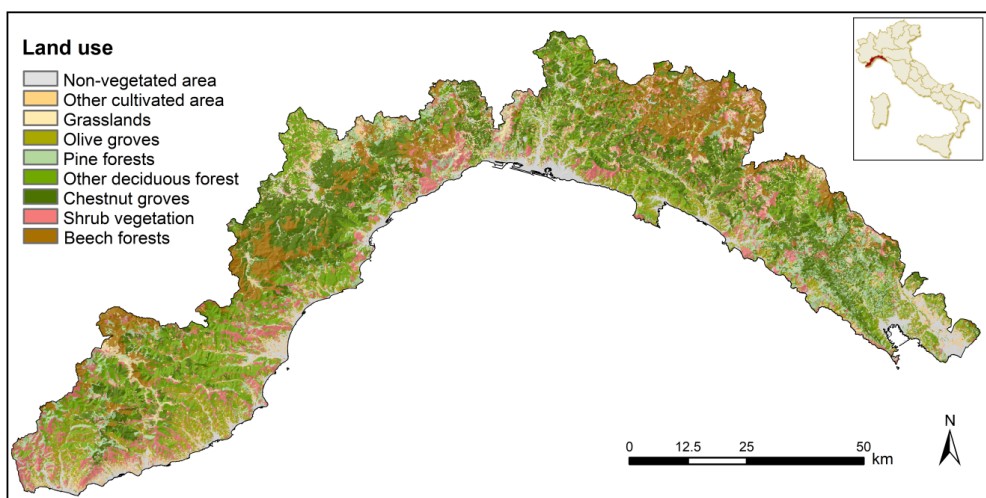

**Figure 1.** Study area.

Forest covers more than 70% of the total surface [42], and it is characterized in large part by the association between beech (*Fagus*) and silver fir (*Abeis alba*). These two wooded species, typical of the ancient forests of the Ligurian Apennines, are currently associated with Scots pine (*Pinus sylvestris*) and larch (*Larix decidua*). In the western and eastern tips of the region, there is a consistent spread of Mediterranean forest with prevalent Aleppo pine (*Pinus halepensis*) and Maritime pine (*Pinus pinaster*), often mixed with numerous broad-leaved trees. The natural populations of these two species, in particular for the maritime pine, are very localized, as most of these pine forests are of artificial origin, with ages ranging between 50 and 120 years [43]. In addition, agricultural and grazing activities have progressively been abandoned in recent decades, favoring the occurrence of wildfire events, which can spread rapidly using the highly flammable pines and shrub species.

The spatial configuration of the territory accounts for a mild climate year-round, including average temperatures of 7–10 °C in winter and 23–24 °C in summer. Mountains very close to the coast create an orographic effect, resulting in abundant rainfall, especially during the autumn months. The average value for rainfall is is 900–1300 mm/year. On the eastern coast, the main cities of Genoa and La Spezia can see up to 2000 mm of rain in a year.

Wildfires in Liguria are recurrent throughout the year. They occur primarily in July–September and February–March. The main causes of the spatial and temporal distributions of wildfires in this region are the following: (i) the climate, which is characterized by long dry period in summer, especially along the coast, but also in winter, caused by dry winds from the north, which blow over herbaceous cover in curing status; (ii) the topography of the area, which favors the fire spreading along the forested steep slopes; (iii) the heterogeneous vegetation, characterized by high percentages of forest canopy; (iv) the human pressure, in terms the rural exodus, urbanization, a growing road network, and agrocultural, forestry, and pastoral activities that followed World War II and that greatly extended the wildland–urban interface [44,45].

### 2.2. Wildfires Dataset

The input dataset used in the present study is the same one used in [15], consisting of mapped burned areas spanning a 21-year period (1997–2017) (Figure 2), provided by the regional forestry service [46]. This wildfire inventory was created based on GPS-surveying and subsequent digitalization over the cadastral map (scale 1:10,000) and includes the starting date of each event. The availability in Liguria of a mapping of fire perimeters (through ground measurements) allows one to carry out a thorough analysis that is not affected by the drawbacks of satellite-based digital observations. The final dataset comprises 8217 wildfires, with an annual mean of 391 fires, burning an area of about 3035 ha per year (Figure 2). To account for the fires' seasonality, two macro seasons have been considered:

winter, from November to April, and summer, from May to October. These two seasons are characterized by a quite distinct spatial distributions of the burned areas (Figure 3) and were therefore analyzed separately.

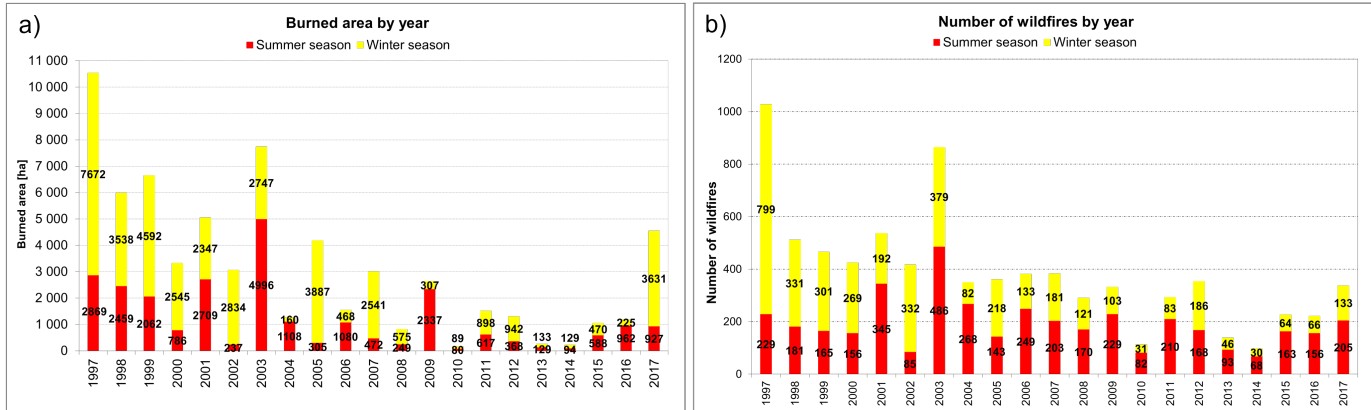

**Figure 2.** Yearly burned area (**a**) and number of fires (**b**) in Liguria during the summer and winter in the period 1997–2017.

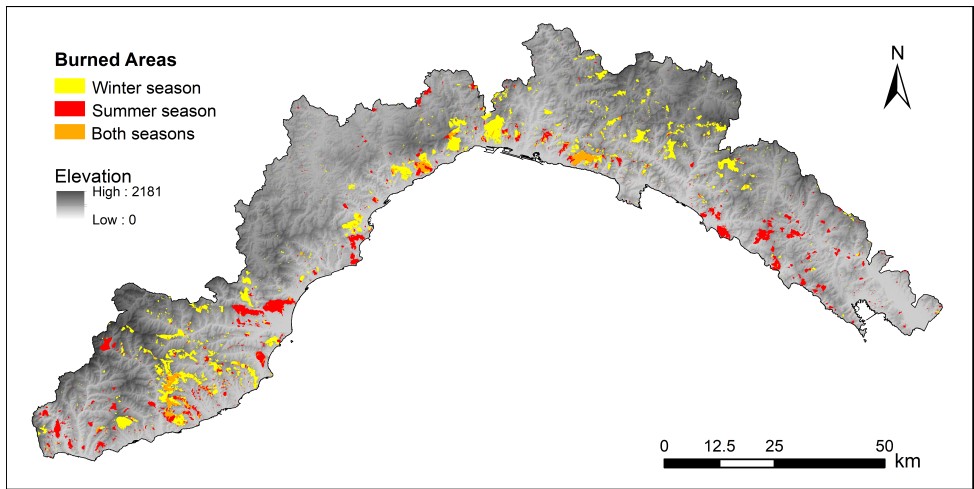

**Figure 3.** Burned area dataset for summer and winter.

### 2.3. Predictor Variables

The predictor variables describe the intrinsic characteristics of the territory and provide information on the associated dependent variable (i.e., the burned areas) with regard to the susceptibility of an area to wildfires. Predictors can be divided into two main groups: the first providing information on the topography and land cover (geo-environmental features) and the second related to anthropogenic features. The first group includes the digital elevation model (DEM) and derived features (slope, northness, and eastness), the type of vegetation (local pixel information), and the neighboring vegetation (within the surrounding pixels). The second group includes the Euclidean distances to an anthropogenic features (i.e., urban areas, road networks, pathways, and crops), and if a pixel belongs or not to a protected area. All this digital information has been provided by the Authority of the Liguria Region and is available at the official geo-portal (https://geoportal.regione.liguria.it/, accessed on 17 November 2022) [46]; a detailed description can be found in [15]. All the spatial layers were rasterized, re-sampled, and spatially aligned to match the same reference raster with a spatial resolution of 100 m.

In the present study, we used an updated version of the map of vegetation types. Information on land use came from the Regional Land Use map (scale 1:10,000), which was

improved for what concerns the class "forest". Namely, the forest areas were reclassified according to the Regional Forest Type map (scale 1:25,000), which provides more detailed information on the different sub-classes. The final map included more than one hundred classes, finally re-aggregated into 37 main types. Specifically, the non-vegetated areas were grouped into a unique class named "non-flammable area", because they are not the object of wildfire events.

Moreover, all the areas classified as "burned area" were reclassified based on the classes of vegetation that covered these areas before the wildfires, evaluated by expert knowledge and by means of maps dated back to an antecedent period. This allowed us to create an accurate map holding detailed information on the different classes of vegetation, especially as concerns the forest cover types. This information had two functions in the present work: (1) the vegetation map has been used as a categorical variable, to describe the class of vegetation as local information characterizing each single pixel (referred to "local vegetation"); (2) for each pixel, a Moore neighborhood of order 2 (resulting in 24 surrounding pixels) was evaluated (Figure 4), allowing us to estimate the percentage of each class of vegetation or non-flammable area within the neighboring pixels (referred to as "neighboring vegetation") and resulting in 38 additional numerical predictor variables (Table 1).

**Table 1.** List of the predictor variables adopted by the different ML models implemented in this work.

| Variable Group | Variable Name | Type | Unit of Measure | Model |
|---|---|---|---|---|
| Topographic | Elevation | Continuous | [m] | All |
| | Slope | Continuous | [°] | All |
| | Northing | Continuous | - | All |
| | Easting | Continuous | - | All |
| Anthropic | Distance from urban areas | Continuous | [m] | All |
| | Distance from Crops | Continuous | [m] | All |
| | Distance from Roads | Continuous | [m] | All |
| | Distance from Tracks | Continuous | [m] | All |
| Vegetational | Vegetation (local) | Categorical (30 cat.) | - | RF Global Vegetation, RF local vegetation |
| | Neighboring vegetation (30 variables) | Continuous | [%] | RF Global Vegetation, RF neighbouring vegetation, SVM, MLP |

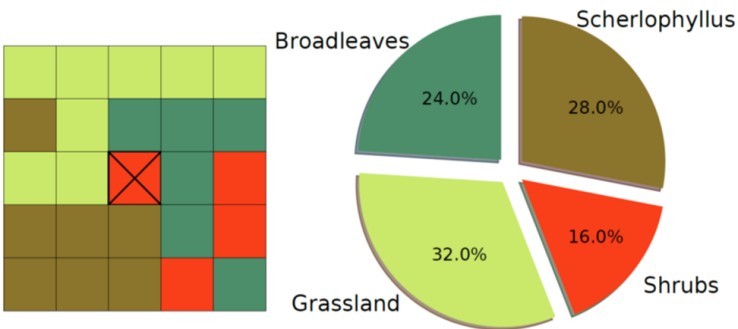

**Figure 4.** Neighboring vegetation variables—computational scheme.

Different variations of ML models can be trained by changing the set of input data. In this work, different strategies were tested for the vegetation part of the input set of predisposing factors. Some models, called in the following "Local Vegetation models", were trained using just the categorical value of vegetation for each pixel. "Neighboring Vegetation models", in turn, were trained using just the 38 continuous variables expressing the percentage of the *i*-th vegetation category in the proximity of the considered pixel ($i \in \{1, 38\}$). In addition, we tested a third ML model category, which included in the

predisposing factor set both the local and the neighboring vegetation layers, referred to "global vegetation". The three models of vegetation were compared using random forest, allowing us to assess the relative importance of the vegetation types as predisposing factors in wildfire prediction.

### 2.4. The Methodological Workflow

The methodological workflow developed in the present study includes the following process:

1. Elaboration of the input dataset: pre-processing of the raster describing the predictor variables (i.e., topographic, anthropogenic, and vegetation features) and the independent variable (i.e., the wildfire dataset).
2. Selection of the testing and training subsets: 3 out of 17 years were randomly selected for the testing subset based on a clustering procedure, to ensure a fair representation of the possible wildfire trends.
   - Selection of the validation subset (via spatial-cross validation): the training subset was then split into 5 parts, and the model was trained on the remaining four parts—the one left out was alternated.
3. Implementation of the machine learning (ML) algorithms, namely, random forest (RF), multi-layer perceptron (MLP), and support vector machine (SVM), for the spatial prediction of wildfire susceptibility.
4. Evaluation of the performance indicators for each ML algorithm and for the two seasons.
   - The AUC (area under the curve) ROC (receiver operating characteristic) were evaluated over the testing dataset.
   - The root mean-square error (RMSE) between the values resulting from the three ML-models and the testing subset was also evaluated.
5. Elaboration of the wildfire susceptibility maps, based of the probabilistic outputs resulting from the three ML implemented models.
6. Assessment of the importance of the predictor variables, obtained by evaluating their rankings and the marginal effect on the predicted outcome.
   - This was achieved with RF, which can handle both numerical variables (e.g., the percentage of neighboring vegetation) and native categorical variables (e.g., the classes of vegetation at the pixel level).

In the following, the three algorithms that were employed for the elaboration of seasonal wildfire susceptibility maps are briefly introduced. The methodology and the experimental settings are described in more detail in the next sections.

### 2.5. Machine-Learning Algorithms

Generally speaking, ML includes algorithms capable of learning from data by modeling the hidden relationships linking a set of input and output variables. The independent input variables are the predictors of the investigated phenomenon (also known as "features"), and the output dependent variables represent the occurrences of the phenomenon (also known as output "labels").

The model is usually evaluated by splitting the input dataset into training, validation, and testing subsets. The training subset allows one to train the model, and the validation subset allows calibrating its parameters. The fitted model can then be used to make predictions with the testing subset. This last set is an independent dataset which is assumed to have the same characteristics as the training data, and it allows one to provide an unbiased evaluation of the model's performance.

The present study deals with making predictions on the probability of an area burning in the future, based on the burned areas observed in the past and on the predictor variables. This can be considered a binary classification problem, seeking to predict whether a given pixel belongs to the class burning (label "one") or unburning (label "zero"). Many ML algorithms support a probabilistic output (ranging continuously from 0 to 1), which

estimates the probability for a pixel to be classified as "burning" or "unburning". This value can be obtained, for example, by normalizing the output over the total number of iterations. ML probabilistic outputs are meaningful to elaborate susceptibility maps, indicating the potential to experience a particular hazard in the future in terms of relative spatial likelihood.

For the purpose of the present study, all the pixels corresponding to burned areas in the wildfire dataset were labeled to "one". To guarantee a balanced representation of the two classes, a number of non-burned pixels (labeled to "zero") equal to the number of pixels labeled "one" was randomly extracted from the non-burned area.

### 2.5.1. Random Forest

Random forest (RF) is an ensemble-learning algorithm for classification and regression based on decision trees [47]. Decision trees are supervised classifiers formed by root nodes and child nodes which can develop at multiple levels. Decisions at the node level are made based on the training predictor variables. At each split, only a subset of these variables are randomly selected to avoid neglecting those that have less influence on the output. The number of variables to be considered (`mtry`) and the number of trees (`ntree`) are the two parameters of the model that need to be optimized. In this study, these parameters were set to the rounded up square-root of the number of predictor factors for `mtry` and 750 for `ntree` following [15]. In order to perform a meaningful split, the prediction error is normally computed on a subset of observations that are not used in the training subsets (called "out-of-bag"—OOB). OOB include about one-third of the testing data, selected by bootstrapping (i.e., random sampling with replacement). At each split of a decision tree, the Gini impurity allows one to determine how the observations should split nodes to form the tree. This step is iterated until each node contains only one observation. For a classification problem, the prediction of new data is finally computed by counting the maximum voting.

The relative importance of each variable can be assessed by evaluating the mean decrease accuracy (MDA). The MDA is estimated by measuring, across all the trees, how much the tree nodes that use a given variable enable reducing the mean-square errors on the OOB. In addition, the partial dependent plot (pdp) gives a graphical depiction of the marginal effect of each variable on the class probability over different ranges of continuous or discrete values. In the present work, it was used to analyze the effect of different vegetation types [48]: for each class, positive values are associated with likely occurrence of the phenomenon (i.e., wildfire), and negative vales indicate its likely absence.

For the computation of RF, we used the the R package `randomForest` and the function `partialPlot` to generate the partial dependence plots [49].

### 2.5.2. Multi-Layer Perceptron

Multi-layer perceptron (MLP) belongs to the class of machine learning algorithms called artificial neural networks (ANN), and is widely used to solve nonlinear data analysis problems. ANN mimics the architecture of the human brain by making use of a set of interconnected neurons. It can be a solid tool for the modeling of problems where relationships between causal predisposing factors and outputs (responses) are complex [23,50]. The human brain is imitated through initiation of a learning process on the available data and storing that knowledge with synaptic weights.

Several neural network structures are available in the literature for different purposes. In this work, the MLP architecture was adopted. According to the definition, three characteristics have to be defined in order to introduce an MLP: a layer of input neurons, a layer of output neurons, and one or more intermediate layers (also called "hidden layers"). Neurons belonging to the same hidden layer are not connected, and every neuron of each layer is connected to every neuron of the adjacent layers. The algorithm is trained via back propagation algorithm (BPA). The idea of BPA is to recalculate the weights array in the last neuron layer based on error (or loss) calculation and proceed towards the previous layers, from back to front. In this way, all the weights in each layer area recalculated, from the last

one until reaching the input layer of the network. The initial weights were set randomly and subsequently iteratively updated by BPA.

For the computation of MLP, we used the function `mlp` implemented in the R package `RSNNS` [51]. The number of epochs and the learning rate of our model were set to 500 and 0.1, and for the other parameters we kept the default values (i.e., one hidden layer of 50 neurons).

### 2.5.3. Support Vector Machine

Support vector machine (SVM), also known as the maximum-margin hyperplane [52], is a ML technique that has largely been applied in the literature to the study of problems of geo-environmental interest [53]. The SVM is based on the statistical learning theory; the input dataset is mapped into a higher-dimensional feature space via non-linear transforms involving kernels. The objective is to find the best separating hyperplane between different output labels [54]. The classifier is a linear hyperplane in the transformed feature space, though it may be non-linear in the original input space, due to the kernel-based non-linear transform.

For instance, considering a training dataset of each label pairs $(X_i, y_i)$, where $i = 1, 2 \cdots, N$, $x_i$ is, for example, the input vectors of the wildfire-predisposing factors, $y_i$ is the label associated with the input vector $x_i$ (that is, "burning" or "non-burning"), and $N$ is the size of the training dataset. In principle, $y_i$ can vary in any set, e.g., $\{0, 1\}$, but most computational implementations retain the choice $y \in \{1, -1\}$ [55]. The SVM model will search an hyper-plane which separates between the two classes ("burning" and "non-burning"), maximizing their gap.

The kernel function of the radial basis function (RBF) is the most commonly used to model natural hazard susceptibility [23,54]. In formulas, it reads:

$$K(\mathbf{x}_i, \mathbf{x}_i) = exp\left(-\gamma \|\|\mathbf{x}_i - \mathbf{x}_j\|\|^2\right), \quad \gamma > 0 \tag{1}$$

The kernel width ($\gamma$) [26] controls the extent of the training sample's influence on the model predictions, affecting the susceptibility output maps produced by the SVM using such kernel. Low values for kernel width express a long-range influence, and high values represent a localized one. The kernel width $\gamma$ balances between bias and variance errors. In the involved minimization problem, the regularization parameter also needs to be tuned.

For the computation of SVM, we used the function `ksvm` implemented in the R package Kernlab [56].

Both SVM and MLP were used by relying on the wrapper and ML process optimizer package `rminer` [57].

### 2.6. Model Evaluation

2.6.1. Spatial Cross-Validation

The validation subset, allowing one to calibrate the parameters of a ML algorithm, is often selected randomly from the training input data. However, in geo-environmental modeling, the random selection can cause the selection of observations falling near to each other in the two subsets, leading to an over-estimation of the predictive performance of the model. This circumstance is known as "spatial auto-correlation", grounded on the fact that close features generally have similar characteristics. To overcome this issue, training and validation subsets were selected far enough apart in the geographic space, by adopting the spatial $k$-fold cross validation with $k = 5$ [15]. This consists of splitting the original training data into $k$ folds, removing a fold at a time, training the model on the remaining $k - 1$ folds, and validating the model using the fold that has been left out. The process is iterated $k$ times, and the different evaluation scores resulting from each folding are then averaged to produce the final results. In the present study, the space was organized into spatial blocks of 15 × 15 km, resulting in 50 blocks covering the entire study area, where each single fold included 10 blocks (corresponding to the 20% of the observations).

### 2.6.2. Selection of the Testing Subset

In this work, the concept of "model generalization" has been widened to include not only the spatial, but also the temporal dimension. This represents a novel contribution to the classical validation procedure in ML. Indeed, ML models are often evaluated considering only the validation subset (i.e., cross-validation or out-of-bag for random forest), which contains not completely unused observations and can result in a over-fitting. At best, for a spatio-temporal dataset, the testing subset is selected considering the last few years of observations [15,36], which makes sense when evaluating the ability of the model to make good predictions in the near future. Nevertheless, this procedure neglects the spatial and temporal auto-correlation among the observations (i.e., the training subset) and predictions on new data (i.e., the testing subset), meaning that observations close to each other, both in space and in time, tend to have similar characteristics. In addition, it can happen that the last years of observations have different characteristics compared to the training subset because of a change in the surrounding conditions, and thus these data could be more suited to being included in the testing subset.

To overcome this issue, the testing subset was implemented by selecting, for both winter and summer, a set of years that are fully representative of the possible wildfire trends observed throughout the entire area and study period. This was obtained by splitting the original input wildfire dataset into three different clusters using the K-means algorithm. The idea was to find homogeneous sub-groups with similar characteristics and tendencies within each group, while maximizing the difference among groups. As clustering criteria (i.e., the attributes of similarity among members of the same cluster), we considered the total number of wildfires and the average burned area per year; see Figure 5. Then, for each one of the three clusters, one year was chosen randomly and assigned to the testing subset. As result, the years 2001, 2010, and 2016 were selected for summer, and 2003, 2007, and 2015 for winter to form the testing subsets.

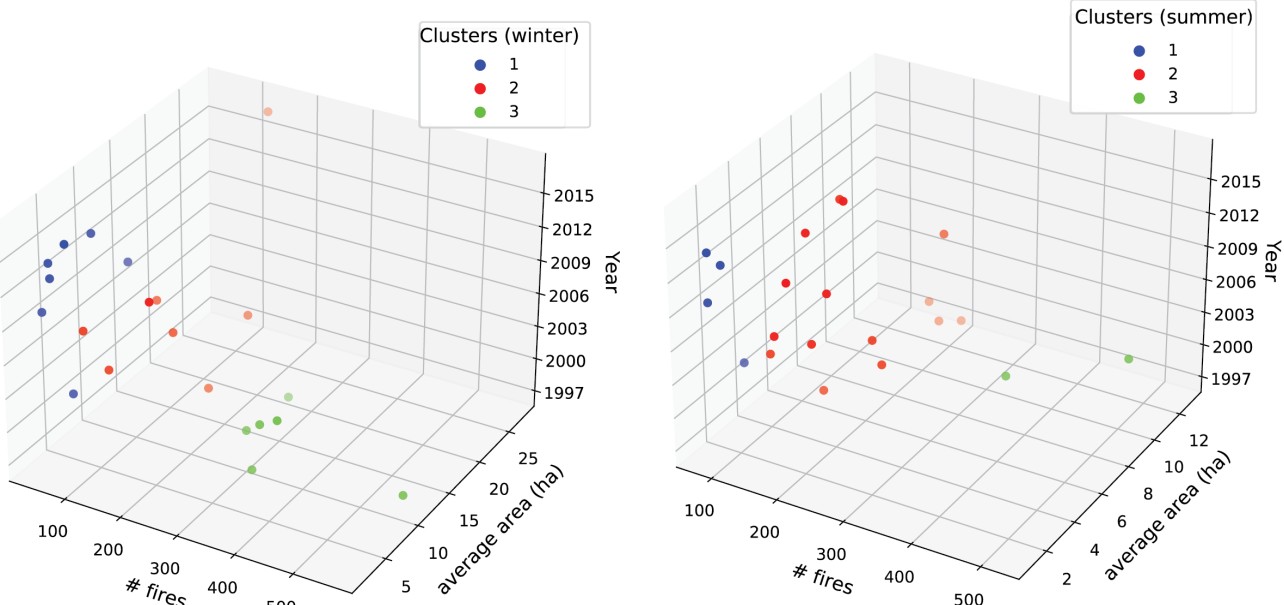

**Figure 5.** Clustering of the wildfire dataset for winter and summer, resulting from K-means (with K = 3).

### 2.6.3. Performance Metrics

The performance of a predictive model can be evaluated by predicting the results over previously unused data—that is, the testing dataset. The area under the receiver operating characteristic (ROC) curve (AUC) is an evaluation score broadly used as an indicator of the goodness of the model in classifying areas that are susceptible to a certain hazard.

The ROC curve is a graphical technique based on the plotting of the percentage of correct classification (the true positives rate) against the false positive rate (occurring when an outcome is incorrectly predicted as belonging to class "one" when it actually belongs to class "zero"), evaluated for many thresholds. The AUC value lies between 0.5, denoting a bad classifier, and 1, denoting an excellent classifier.

As an additional indicator of the algorithm's goodness we computed the root mean-square error (RMSE), based on the difference between the predicted outputs, expressed as a probability value in the interval $[0, 1]$, and the observations, which assumed the value "one" for burning area and "zero" for unburned areas. Therefore, for a model performing good predictions (i.e., model outputs for burning areas $\in \{0.5, 1\}$ and for unburned areas $\in \{0, 0.5\}$), we can expect an RMSE value lying between 0 and 0.5.

Finally, the performances of the three algorithms (RF, MLP, and SVM) were further assessed and compared by calculating the percentage of an area that fell within a burned area in the testing subset and then looking at its probabilistic predicted value. Output values were split into five ranges (25th, 50th, 75th, 90th, and 95th). The 25th percentile indicates the 25% of the area with the lowest probabilistic predicted values of burning, while the 95th percentile indicates the 5% of the area with the highest ones, and so on for the intermediate ranges.

## 3. Results and Discussion

### 3.1. Comparison of the Three ML Algorithms

The main output resulting from the three ML algorithms is the probabilistic predicted value for each pixel to burn under the influence of the geo-environmental predisposing variables. Specifically, we compared RF, MLP, and SVM under the same conditions, which, to restate, were the following: (i) 5-fold cross validation, to ensure good generalization of the results and avoid over-fitting; (ii) a set of numerical variables (neighboring vegetation; DEM; slope; northness; eastness; belonging or not to a protected areas; distances to urban areas, road networks, pathways, and crops) that can be handled by all the three algorithms.

RF resulted to be the best predictive model, both for winter and for summer (Figure 6), as attested by the highest values of AUC (0.944 and 0.953, respectively), followed by MLP (0.921 and 9.940), and lastly, by SVM (0.916 and 0.931). The RMSE estimator confirmed this result; see Table 2. Indeed, RF has the lower values for both the seasons (0.329 in summer and 0.335 in winter) compared to MLP and SVM, which have higher values for summer (0.344 and 0.358, respectively) and winter (0.353 and 0.360, respectively).

**Table 2.** RMSE and AUC values for testing burned area, for both winter and summer, and for all the implemented modes.

| Winter | | RMSE | AUC |
|---|---|---|---|
| | Neighboring Vegetation | 0.335 | 0.944 |
| Random Forest | Local vegetation | 0.367 | 0.906 |
| | Global vegetation | 0.342 | 0.939 |
| SVM | Neighboring Vegetation | 0.36 | 0.916 |
| MLP | Neighboring Vegetation | 0.353 | 0.921 |
| **Summer** | | **RMSE** | **AUC** |
| | Neighboring Vegetation | 0.329 | 0.953 |
| Random Forest | Local Vegetation | 0.37 | 0.911 |
| | Global vegetation | 0.328 | 0.952 |
| SVM | Neighboring Vegetation | 0.358 | 0.931 |
| MLP | Neighboring Vegetation | 0.344 | 0.94 |

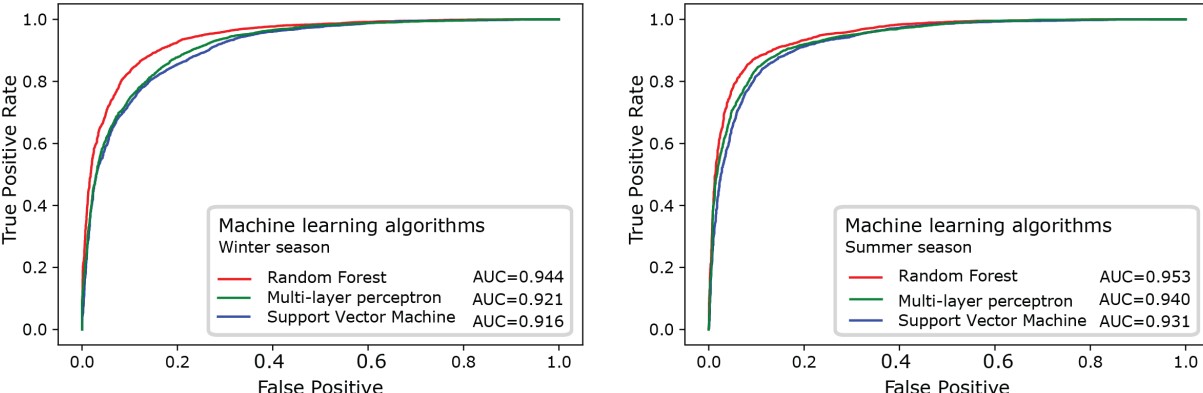

**Figure 6.** ROC curves for the three ML algorithms with the corresponding AUC values.

Table 3 shows the percentage of burned area in the testing dataset (Testing BA) within each percentile class range (Classes) and the corresponding probabilistic predicted value (Prob. Value). The same percentile range's limits can correspond to different values of the probabilistic output, allowing the competent authority the make decisions based on a given threshold for the highly susceptible area rather than on the raw output values. A model that predicts good results is expected to produce a higher percentage of the testing burned area within the highest percentile ranges and a lower percentage in the lowest ranges. Indeed, this is exactly what happened with the three algorithms (RF, MLP, and SVM) for both the seasons. Moreover, looking at the 25% of the area with the highest probability of burning (corresponding to the class >75%), results indicate that RF performs better that SVM and MLP: 93.65% of the test set's burned area was predicted for winter and 94.80% for summer, compared to 89.57% and 87.30% for winter and 92.68% and 91.93% for summer for MLP and SVM, respectively. Likewise, given the area with the lowest probability of burning (corresponding to the class 25%), the three algorithms allocated a very low extent of test burned area for winter (0.27%, 0.48%, and 0.42% for RF, MLP, and SVM, respectively) and for summer (0.18%, 0.10%, and 0.34% for RF, MLP, and SVM, respectively).

**Table 3.** Percentages of burned area in the testing subset belonging to different output percentile class ranges for SVM, MLP, and RF with neighboring vegetation. The last line for each season represents the top 25%.

| Winter Season | | SVM | | MLP | | RF | |
|---|---|---|---|---|---|---|---|
| Classes | Total Area (%) | Testing BA | Prob. Value | Testing BA | Prob. Value | Testing BA | Prob. Value |
| 25% | 25 | 0.42 | 0.13 | 0.48 | 0.13 | 0.27 | 0.10 |
| 50% | 25 | 2.14 | 0.22 | 1.55 | 0.25 | 1.43 | 0.21 |
| 75% | 25 | 10.14 | 0.46 | 8.43 | 0.46 | 4.65 | 0.41 |
| 90% | 15 | 19.97 | 0.74 | 21.67 | 0.68 | 17.67 | 0.67 |
| 95% | 5 | 19.05 | 0.85 | 18.57 | 0.81 | 17.40 | 0.81 |
| 100% | 5 | 48.27 | 0.99 | 49.30 | 0.99 | 58.58 | 1.00 |
| >75% | 25 | 87.30 | | 89.54 | | 93.65 | |
| **Summer Season** | | **SVM** | | **MLP** | | **RF** | |
| Classes | Total Area (%) | Testing BA | Prob. Value | Testing BA | Prob. Value | Testing BA | Prob. Value |
| 25% | 25 | 0.34 | 0.09 | 0.10 | 0.08 | 0.18 | 0.05 |
| 50% | 25 | 1.11 | 0.17 | 1.35 | 0.21 | 0.83 | 0.18 |
| 75% | 25 | 6.62 | 0.50 | 5.87 | 0.47 | 4.14 | 0.45 |
| 90% | 15 | 15.99 | 0.77 | 13.82 | 0.69 | 10.22 | 0.69 |
| 95% | 5 | 19.79 | 0.83 | 16.82 | 0.82 | 14.64 | 0.81 |
| 100% | 5 | 56.14 | 0.99 | 62.04 | 1.00 | 69.94 | 1.00 |
| >75% | 25 | 91.93 | | 92.68 | | 94.80 | |

### 3.2. Susceptibility Maps

Wildfires susceptibility maps were elaborated based on the probabilistic predicted values resulting from the three ML-based approaches (Figure 7) and classified using the same ranges as in Table 3. The implementation of RF, MLP, and SMV in the model resulted in quite similar maps. Areas of high/very high and low/very low susceptibility classes were detected in the same locations in the three cases, for both the seasons. In winter, the very high susceptibility class (above the 90th percentile) is mainly located in the more elevated inland areas, and in summer it is mainly distributed in the coastal area. This spatial pattern can be attributable to the state of the vegetation, which is more burnable at higher altitudes in winter and at lower elevations and closer to the coast in summer. This characteristic is a consequence of the senescence of the vegetation in the mountainous areas in winter, while in summer the high temperature and the dry weather cause it the be dryer and more venerable to fires in the plain and close to the touristic areas.

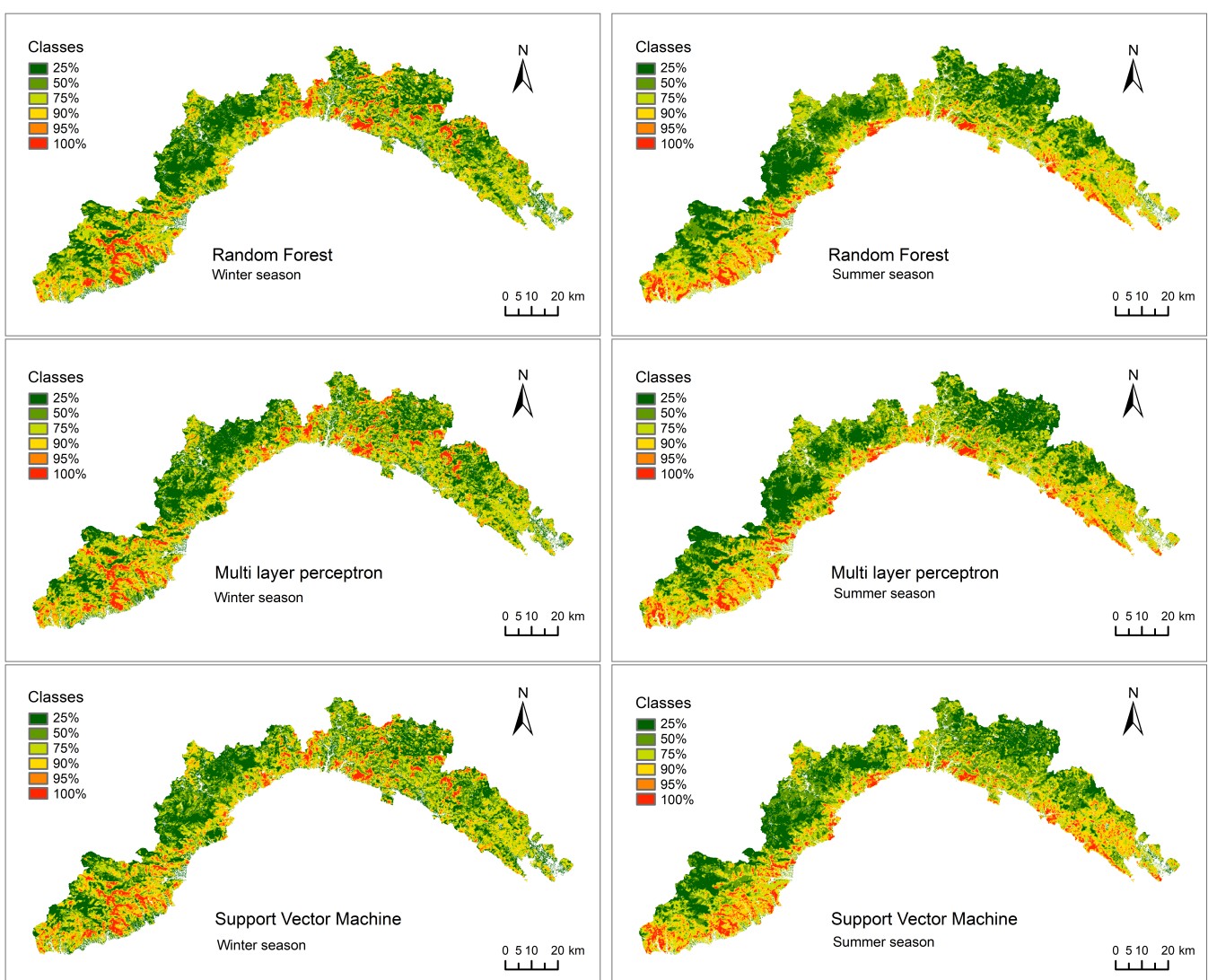

**Figure 7.** Susceptibility maps.

### 3.3. Assessment of the Predictor Variables

#### 3.3.1. Effect of the Neighboring Vegetation

As explained in the Methods, in the present study the type of vegetation was evaluated both as categorical variable, to describe the class of vegetation as "local information" characterizing each single pixel, and as "neighboring vegetation", assessed by estimating

the percentage of each class of vegetation, or non-flammable area, within the neighboring pixels. This led to the creation of a set of additional variables ranging from 0 to 100% corresponding to each class of vegetation. Moreover, a third "global vegetation" model, including both the local and the neighboring vegetation, was implemented.

To evaluate the predictive performances of these three vegetation models and estimate which one gives the best results, they were compared by using RF (Figure 8 and Table 2). We would like to stress again that RF can handle directly both categorical and numerical variables. Indeed, with the adopted R implementation (available in randomForest package), there is no need for transforming "categorical" variables into "numerical" variables, thereby limiting the number of predictors. The resulting ROC curves clearly show that accounting for the "neighboring vegetation" allows one to enhance the performance of the model (see Figure 8), resulting in increasing values of AUC from 0.906 to 0.944 in winter and from 0.911 to 0.953 in summer. The "global model", which includes both the local and the neighboring vegetation, has a similar performance to the neighboring model (AUC equal to 0.939 in winter and 0.952 in summer) (Figure 8) and was considered for the final assessment of the importance of the predictor variables.

As additional validation, Table 4 expresses the quantile-analysis of the burned area, analogously to Table 3. The three RF models performed well in this analysis, but the model without neighboring vegetation had lower performances (the highest percentile class covered 45% of the total BA, while the other choices obtained both more than 55%).

**Table 4.** Percentage of burned area in the testing subset belonging to different output percentile classes for RF with different vegetation models. The last line for each season represents the top 25% percentile.

| Winter Season | | Global Vegetation | | Neighboring Vegetation | | Without Neighboring Vegetation | |
|---|---|---|---|---|---|---|---|
| Classes | Total Area (%) | Testing BA (%) | Prob. Value | Testing BA (%) | Prob. Value | Testing BA (%) | Prob. Value |
| 25% | 25 | 0.34 | 0.09 | 0.27 | 0.10 | 0.94 | 0.12 |
| 50% | 25 | 1.47 | 0.21 | 1.43 | 0.21 | 2.75 | 0.26 |
| 75% | 25 | 4.70 | 0.43 | 4.65 | 0.41 | 8.72 | 0.46 |
| 90% | 15 | 18.09 | 0.68 | 17.67 | 0.67 | 23.79 | 0.70 |
| 95% | 5 | 19.83 | 0.82 | 17.40 | 0.81 | 17.94 | 0.84 |
| 100% | 5 | 55.59 | 1.00 | 58.58 | 1.00 | 45.84 | 1.00 |
| >75% | | 93.50 | | 93.65 | | 87.57 | |
| **Summer Season** | | **Global Vegetation** | | **Neighboring Vegetation** | | **Without Neighboring Vegetation** | |
| Classes | Total Area (%) | Testing BA (%) | Prob. Value | Testing BA (%) | Prob. Value | Testing BA (%) | Prob. Value |
| 25% | 25 | 0.13 | 0.05 | 0.18 | 0.05 | 0.26 | 0.06 |
| 50% | 25 | 1.14 | 0.18 | 0.83 | 0.18 | 2.02 | 0.19 |
| 75% | 25 | 4.04 | 0.46 | 4.14 | 0.45 | 9.50 | 0.53 |
| 90% | 15 | 10.84 | 0.70 | 10.22 | 0.69 | 21.76 | 0.75 |
| 95% | 5 | 14.98 | 0.82 | 14.64 | 0.81 | 17.10 | 0.83 |
| 100% | 5 | 68.85 | 1.00 | 69.94 | 1.00 | 49.34 | 1.00 |
| >75% | | 94.67 | | 94.80 | | 88.20 | |

### 3.3.2. Predictor Variables Importance Ranking

The relative importance of the predictor variables was evaluated with regard to the global vegetation model, based on the value of the mean decrease accuracy (MDA). Higher values of MDA mean that the model strongly benefits from the given predictor to make meaningful choices in assigning a pixel to high or low fire susceptibility to burn. MDA values have been normalized in order to have a sum equal to one and allowing a fear comparison in both the seasons.

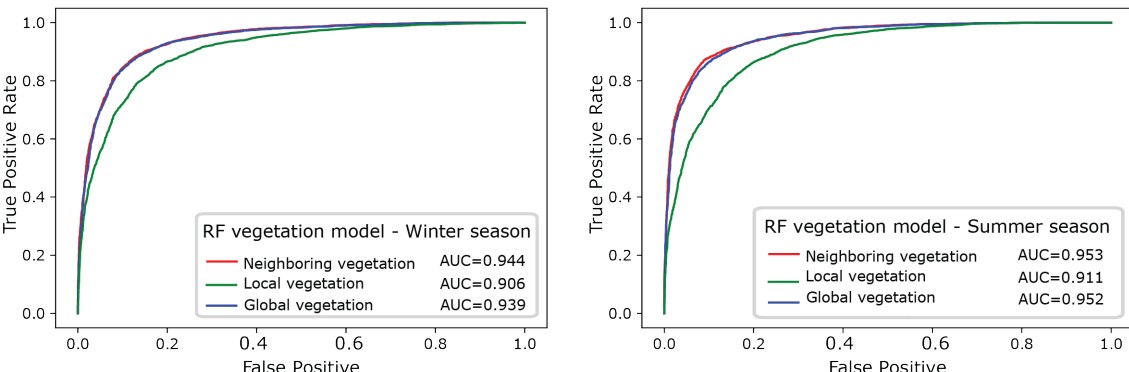

**Figure 8.** ROC curves for the three RF vegetation models and corresponding AUC values.

Vegetation was by far the most important variable, as clearly shown in Figure 9. Here, the local type of vegetation has a normalized MDA of about 0.2 for both the seasons, followed by orographic variables such as elevation and northness. The other variables usually plateau below 0.05. In order to have a clear display, the ranking of the first neighboring vegetation classes are represented in a another plot (Figure 10). In summer, the contributions of moors, shrubs, Mediterranean pines, chestnut trees, beech, hop and hornbeam forests, oaks, and downy oaks to the vegetation mosaic are predominant; and in winter the presence of pastures plays a major role, followed by oaks and moors/shrubs.

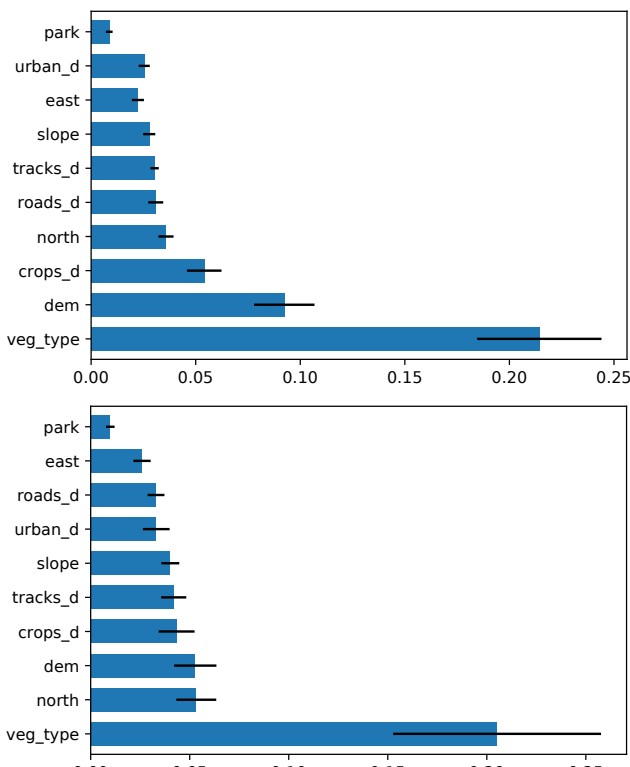

**Figure 9.** Variable importance rankings for RF model (global vegetation), based on the mean decrease accuracy (MDA), of the predictor variables. Summer and winter are portrayed on the top and the bottom, respectively. The blue bars represent the means of the importance score among all the folds, and black lines represent the standard deviation.

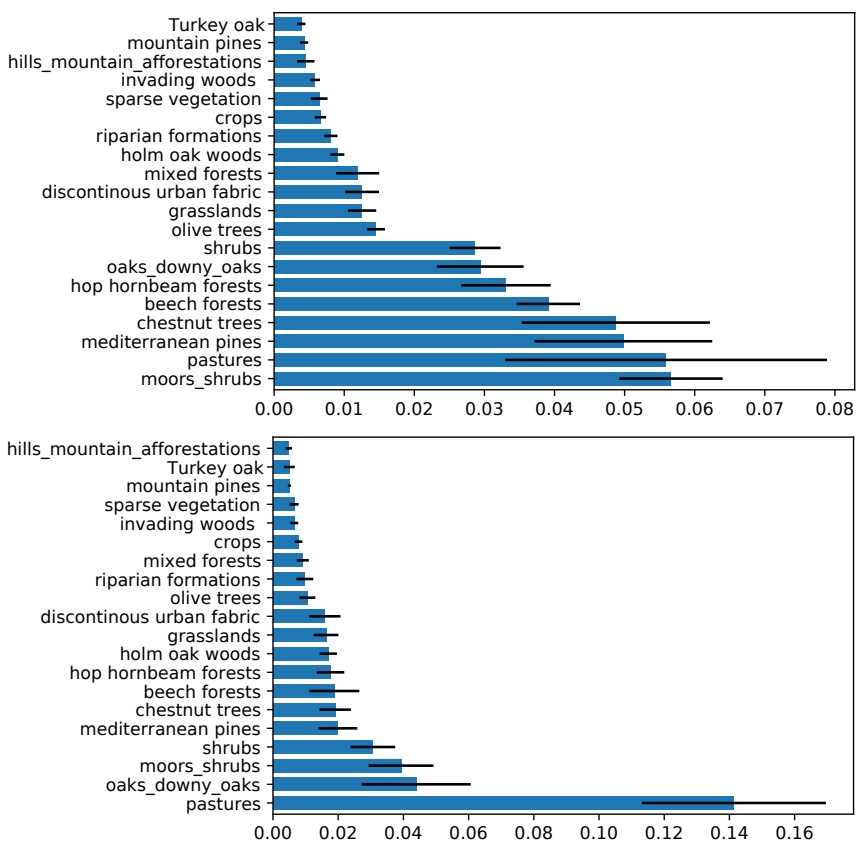

**Figure 10.** Variable importance ranking (based on the MDA) of each neighboring vegetation variable, using the RF model (global vegetation). Summer and Winter are portrayed on the top and the bottom, respectively. The blue bars represent the mean of the importance score among all the folds, and black lines represent the standard deviation.

The partial dependence plot (Figure 11) allows one to infer the magnitude of the relevance of each single vegetation class, and to detect whether a particular type of vegetation is important because it is associated with high or low fire susceptibility. The pixel-related local vegetation categories follow on average similar patterns as their neighboring counterparts. However, it emerges from Figure 11 that, for the summer wildfire susceptibility model, areas with mixtures of crops and natural vegetation, pastures, moors, shrubs, mixed forest, schlerophyllus vegetation, and Mediterranean pines were associated with high levels of fire susceptibility; and beech forests, hop/hornbeam forest, and turkey oak are associated with low levels of susceptibility. For what concerns winter wildfire susceptibility, from Figure 11, it is evident that pastures, mixed forests, moors and shrubs, mixed crops/natural areas, and downy oaks are characterized by high susceptibility; and sessile oaks, beech holm, mountain pines, and Turkey oak are associated with low susceptibility patterns.

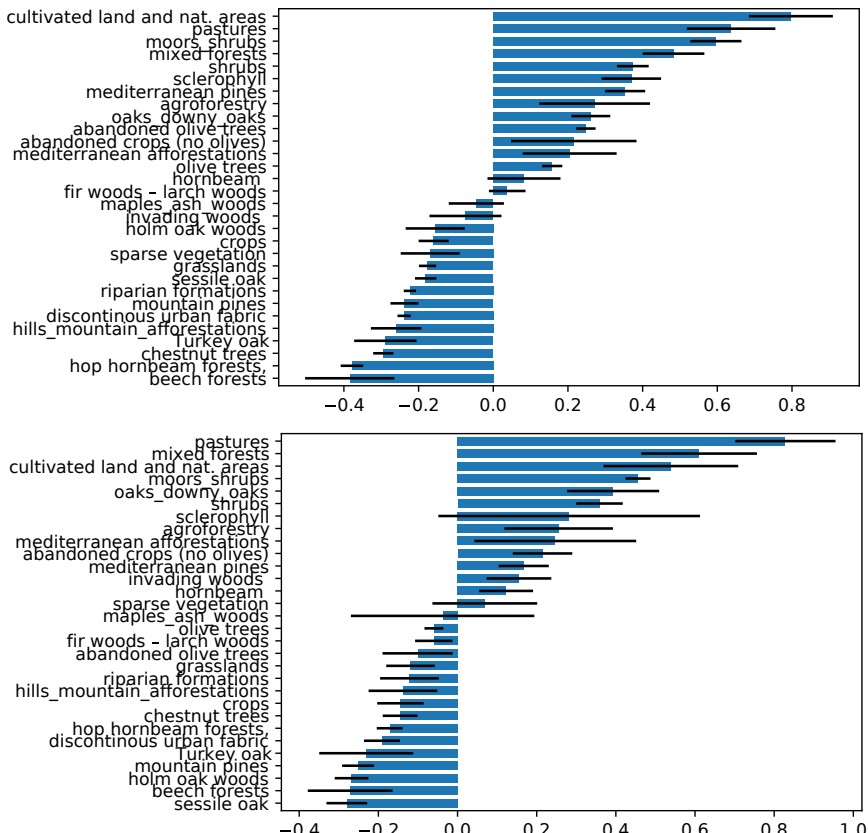

**Figure 11.** Partial dependence plot of the categorical variable corresponding to the local vegetation, for the RF model with global vegetation input set. Summer and winter are portrayed on the top and the bottom, respectively. The blue bars represent the means of the importance score among all the folds, and black lines represent the standard deviation.

## 4. Conclusions

In the present work, we consolidated the use of machine learning to asses the susceptibility to wildfires at the regional scale and to estimate the relative importance of the predictor variables (i.e, land cover, type of vegetation, altitude and its derivatives, nearby infrastructures). In comparison to the previous studies [15,16], the main novelties consisted in the comparison of three algorithms, namely, random forest, multi-layer perceptron, and support vector machine, and in the implementation of an accurate vegetation map as input. This last allowed us to evaluate the impact not only of the vegetation as predictor, but of each single class of vegetation on wildfires' occurrence. Moreover, to account for the temporal and for the spatial variability of the burning seasons (winter and spring), we selected a highly representative testing subset, with the purpose of avoiding the problem of the spatio-temporal auto-correlation with the observations in the training that can cause overfitting.

The main results of the implemented ML models consisted in the probabilistic predicted values finally used to produce susceptibility maps. The spatial pattern distribution of the areas in the different classes of susceptibility did not substantially differ by using RF, MLP, or SVM. Nevertheless, indicators of performance (based on ROC curves and the corresponding values of AUC, and RMS errors and the comparison with the burned area in the testing subset) revealed that RF gives the better performances, followed closely by MLP and SVM.

Finally, we used RF to asses the variables' importance ranking. Indeed this algorithm can handle both numerical variables (such as the percentage of neighbouring vegetation) and native categorical variables (such as the types of vegetation at local pixel level). The most important predictor variable was the vegetation, and a more detailed investigation on

the marginal effect of each type of vegetation allowed us to detect the singles classes that are more or less susceptible to wildfires in summer and winter.

The thorough variable importance assessment carried out in this study showed the importance of explainable ML procedures, shifting from a black box to a more understandable framework. In this specific case, the importance of vegetation cover and its continuity pave the way for further assessment of vegetation impact on wildfire regimes. In addition, it has to be remarked that no climate data were considered as input variables. This is in line with the pioneering work of Tonini et al. [15], and allowed us to shift the focus to the vegetation importance in the wildfire susceptibility patterns of winter and summer. However, the inclusion of climatic variables in the predictors set, e.g., wind, temperatures, precipitation, and soil moisture averages, may shed light on their effects on the wildfire regimes also at a regional level. They can also be useful for depicting climate change scenarios. This is clearly a gap that needs to be filled by researchers in future studies. Moreover, the presented maps are static products that do not consider synoptic data of wind and fuel moisture and need further modeling in order to provide time-varying fire danger maps.

Results of the present work are extremely useful for decision makers in wildfire management and long-term land use planning. Actually, ML-based susceptibility maps are adopted operationally by the Italian Civil Protection in early warning systems for wildfire danger [58–60]. Future works will be devoted to trans-boundary case studies, where susceptibility maps at the macro-regional scale can help in transboundary risk assessment procedures.

In the present study, ML proved to be successful in assessing the susceptibility to wildfires, and also showed how the different topographic, vegetational, and anthropogenic factors affect the propensity of an area to burn. The same approach can be applied to other research domains. Indeed, ML is emerging as a new paradigm in geosciences [61], as it enables us to make sense of the collected data and to discover the hidden relationships between the target events (e.g., natural hazards, geological or atmospheric process, etc.) and the predictor variables.

**Author Contributions:** Conceptualization, A.T., M.T. and P.F.; writing—original draft, A.T. and M.T.; writing—review and editing, A.T. and M.T.; investigation, A.T., S.I., H.I. and M.T.; supervision, P.F. and M.T.; software, A.T., H.I. and S.I.; funding acquisition, P.F. and M.T. All authors have read and agreed to the published version of the manuscript.

**Funding:** This research received no external funding.

**Acknowledgments:** The authors wish to thank Regione Liguria for making available all the data used in the analysis.

**Conflicts of Interest:** The authors declare no conflict of interest.

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
