# Peer review of "Machine-Learning Applications in Geosciences: Comparison of Different Algorithms and Vegetation Classes’ Importance Ranking in Wildfire Susceptibility"

_geosciences, doi:10.3390/geosciences12110424_

Round 1

Reviewer 1 Report

This article analyzes predictions of wildfire susceptibility with three different Machine Learning Algorithms and the role of vegetation in mapping the wildfire susceptibility. The article is well-though out and the results will be of great interest to managers in the area.

 I am curious, though, if there were any prescribed fires in the region in the 21 years of data inventory? If so, how did the authors differentiate between wildfires and prescribed fires? Or this analysis only included wildfires. Also, can the authors calculate percentage of wildfires in each vegetation category? From the results it appears that the variable pasture was at the top of the MDA graphs, which suggests that this variable is quite important in predicting the occurrence of a wildfire. However, if these fires are prescribed fires started by land owners or managers, then the susceptibility maps could be skewed in favor of pastures, and, at least for pastures, would show not weather or not an area would burn, but weather or not a land owner would burn the pasture. Also, can the MLP and SVM methods produce MDA graphs or similar graphs that would depict the importance of different variables in the final model? It would be interesting to see if the variable importance (and their order) would be the same as for the RF model.

 Below I have a few minor comments, which I am providing line-by-line.

 Line 2: Replace “Authors” to “the authors”.

Lines 6-7: To make the sentence clearer, I suggest replacing “we compared Random Forest with Multi-layer Perceptron and Support Vector Machine, to estimate their prediction capabilities” to “we compared models based on Random Forest, Multi-layer Perceptron, and Support Vector Machine, to estimate their prediction capabilities”. Or something similar.

Line 14: Replace “than” to “then”.

Lines 23-24: I argue that the loss of life is a more severe consequence of wildfire than desertification/deforestation. Please check the original citation [3] to make sure that this is what the authors intended to write in the original citation.

Line 27: Replace “and cause every year substantial damages” to “and cause substantial damages every year”.

Lines 31-32: …”with the exception of the unusual year 2017”, when what happened? Why was 2017 unusual? I assume there were more fires that burned in 2017 compared to the decade decreasing average, but it would be helpful if stated clearly.

Line 40: “with a limited long-term positive effect”. What does this mean?

Lines 47-50: Replace “we refer to "susceptibility" to define area” with “we define "susceptibility" as areas…”.

Line 58: Replace “pas” to “past”

Line 60: Replace “for widen” to “to widen”

Line 91: Replace “Authors” to “the authors”. Or simply say “In the present work, we continue…”. I understand avoiding first person pronouns (I/We) to avoid subjectivity; however, it can make science ambiguous and unaccountable (e.g.: “Data was analyzed” vs. “We analyzed the data”; “The soil samples were collected” vs. “We collected soil samples”, etc. I think it’s fine in this context for the authors to use first person pronouns.

Line 94: Replace “elaborate” with a better word. Perhaps “create”?

Line 100: Why weren’t meant temperature and precipitation used in the current study, too?

Line 129: pine Aleppo and Maritime pine are written italicized but these are not the scientific names. Perhaps write in normal font and add the scientific names in italic and in parentheses as for the Scots pine and larch.  

Lines 138-139: replace “with average values approaching the 900-1300 mm/year” with “ranging between 900-1300 mm/year”

Line 145: Unclear

Line 146: I assume the authors meant to say that the spatial and temporal distribution of wildfires is not because of “the topographic assessment of the area” but “the topography of the area”. If so, please correct.

Line 147: Remove “a”

Line 169: Replace “firs” to “first”

Line 191: By elaborate, do the authors mean “create”?

Line 193: Replace “maps” to “map”

Line 199: It is not clear at this stage that the “local vegetation” and “neighboring vegetation” were two models. So perhaps the authors would like to rewrite parts of this paragraph to make it clear that there were three model tested with different vegetation variables.

Line 210: Perhaps the word is “fair”?

Line 266: Write (mtry) in parentheses like (ntree)

Line 351: Replace “non” to “not”

Line 358: I assume “Nerveless” should be “Nevertheless”?

Line 363: “Profitably” implies financial gain. Perhaps the authors would like to use a different word? Maybe “more suited to be included in the …” Or something similar.

Line 366: Replace “is” to “are”.

Line 386-390: I do not understand how RMSE can be used to predict a binary outcome? What would even be the unit of RMSE in this context?

Line 404: Remove the first closing parenthesis, after fitting.

Line 461: Replace “that” to “as”

Lines 483-494: It is interesting to see that the pastures turn out to be at the top of variable importance for both summer and winter predictions. Are there any practices employed by managers/farmers in the region of burning pastures to increase productivity? I believe this practice is used in many regions in Europe but I am curious if this is also the case in the Liguria region? Or are pastures more flammable than forests?

 Figure 1, Figure 2, Figure 3 are identical to figures in the 2020 Paper [citation 12]. This needs to be sorted out with the Journal and see if this is acceptable in terms of copy-rights, despite both manuscripts being published in the same Journal. At a minimum, it should be stated in the figure caption where the figures are taken from.

 Figure. 1. Legend. Is it important to mention Land use based on vegetation type?

Figure 5: Is not mentioned in the text.

Figure 7: Replace “percepron” to “perceptron” for both winter and summer plots.

 Table 2: What is the unit of RMSE. Usually it is computed when predicting categorical data and has the units of the observed data. But in this case the prediction is binary (0/1) so I am not sure what the unit of RMSE is?

 Citations:

6. Author’s family names? Also, is year 2020 or 2021?

54: Author’s family names

55: Author’s family names?

Author Response

Dear Reviewer #1,

Thanks for the time and effort put in this review. We put the response to your questions in the attached PDF, at the section "Reviewer #1".

Best regards

The Authors

Reviewer 2 Report

Reviewer’s Report on the manuscript entitled:

Comparison of different Machine Learning Algorithms to assess the importance of the Vegetation in Wildfire Susceptibility Mapping

The authors compared different machine learning methods for evaluating their capabilities for wildfire susceptibility mapping and for finding the most influential parameters. The topic and results are generally interesting and the manuscript is well-written, but the presentation can be improved. Below, please see my comments:

Line 2. What do you mean by “Authors”? Please replace “Authors” with “we” if it refers to you. Please rephrase this sentence entirely. Similarly for line 91.

Line 14. Please replace “than” with “then”

Line 72. More recent articles can be included and described here:

For example, Salavati et al. [https://doi.org/10.3390/su14073881] applied statistical methods on influential factors, such as climate factors, topographic variables, land use/land cover, distance from road and rivers to generate wildfire risk maps.

Novo et al. [https://doi.org/10.3390/rs12223705] developed a method to automatically find main factors involved in forest fire risk map

Also, please consider breaking this large paragraph (lines 68-90) into two.

Figures 1 and 3: legends should be enlarged. Generally, the font size of the text and numbers in the figures should be well readable and enlarged to have a size approximately the same as figure caption.  The font size of text and numbers in Figure 2 should also be enlarged.

Section 2. Study area is usually in Materials and Methods. So, it is recommended to have Materials and Methods as section 2 and then section 2.1 would be Study area and Section 2.2 datasets and Section 2.3 methods.  Then Section 3 Results, etc. I leave this to the authors to decide.

Line 319. Please explain why y in {-1,1} and not like in line 249 that you said {0,1}. Burning labelled 1 and non-burning labeled 0. Cannot {-1,1} be {0,1}? Please clarify.

Line 351. Please replace “non” with “not”

Table 5. Why local and global vegetation are not considered in SVM and MLP?

The limitations of this study should be mentioned in the Conclusion. As you pointed out in line 145, wind is one of the main reasons of wildfire spread. Soil moisture and humidity can also be other factors for fire spread and risk forecasting which I think are not considered in your study, so they can be mentioned as limitations.

Thank you!

Author Response

Dear Reviewer #2,

We thank you for the efforts put in  reviewing our manuscript.
We provide a point-by-point response to your comments in the attached PDF document, in the section "Response to Reviewer #2".

Best regards

The Authors
